# Reconstruction and normalization of LISA for spatial analysis

**Yanguang Chen** *

Department of Geography, College of Urban and Environmental Sciences, Peking University, Beijing, China

* chenyg@pku.edu.cn

## Abstract

The local indicators of spatial association (LISA) are important measures for spatial autocorrelation analysis. However, there is an inadvertent fault in the mathematical processes of deriving LISA in literature so that the local Moran and Geary indicators do not satisfy the second basic requirement for LISA: *the sum of the local indicators is proportional to a global indicator*. This paper aims at reconstructing the calculation formulae of the local Moran indexes and Geary coefficients through mathematical derivation and empirical evidence. Two sets of LISAs were clarified by new mathematical reasoning. One set of LISAs is based on non-normalized weights and non-centralized variable (MI1 and GC1), and the other set is based on row normalized weights and standardized variable (MI2 and GC2). The results show that the first set of LISAs satisfy the above-mentioned second requirement, but the second the set cannot. Then, the third set of LISA was proposed and can be treated as canonical forms (MI3 and GC3). This set of LISAs satisfies the second requirement. The observational data of city population and traffic mileage in Beijing-Tianjin-Hebei region of China were employed to verify the theoretical results. This study helps to clarify the misunderstandings about LISAs in the field of geospatial analysis.

## 1 Introduction

Geography has two core concepts on location effect: difference and dependence. The former is related to a classical topic of geography, while the latter is related to spatial correlation analysis. The concept of spatial difference is also termed regional differences, which came from areal differentiation [1–3]. The traditional concept of difference seems to be in contradiction with the pursuit of general laws, so geography embarks on the road of "exceptionalism" [4]. After the quantitative revolution (1953–1976), geography began to attach importance to spatial organization and correlation, which indicates spatial dependence. Spatial interaction models and spatial autocorrelation analysis are the main approaches to research spatial correlation processes [5, 6]. Spatial autocorrelation is originally a biological statistic concept, which is mainly used to evaluate whether the spatial sampling results meet the traditional statistical requirements [7–9]. When geographers introduced spatial autocorrelation measure into geospatial analysis, they found that there are few spatial uncorrelated phenomena. In this context, the spatial autocorrelation analysis method was developed [10–12]. The early spatial

**Data Availability Statement:** The data underlying the results presented in the study are available from the supporting information files.

**Funding:** The project is funded by the National Natural Science Foundation of China (42171192). The funders had no role in study design, data

collection and analysis, decision to publish, or preparation of the manuscript.

**Competing interests:** The authors have declared that no competing interests exist.

autocorrelation analysis was only at the global level, rarely involving the local level, so it provided limited geospatial information. In other words, the initial spatial autocorrelation focuses on spatial dependence rather than spatial difference. After the theoretical revolution in the later period of the quantitative revolution was frustrated, the traditional regional trend of thought of geography returned quietly, and the concept of regional difference was again valued by geographers with a new expression of spatial heterogeneity [13]. Tobler proposed the first law of geography based on spatial dependence [14], and Harvey proposed that spatial heterogeneity be the second law of geography [15]. The study of spatial heterogeneity naturally involves spatial locality. According to Fotheringham [16–18], there are three trends in the development of quantitative geography: localization, computation and visualization. In this sense, local spatial autocorrelation analysis came into being [13, 19–22]. Therefore, spatial difference (heterogeneity) and spatial correlation (dependency) have reached the same goal through different routes [13, 23].

Local spatial autocorrelation analysis is developed on the basis of global spatial autocorrelation analysis. The Local Indicators of Spatial Association (LISA) proposed by Anselin [19] plays an important role in the local correlation analysis of geographical research. LISA includes local Moran indexes and local Geary coefficients. These spatial statistics, together with the *G* index proposed by Getis and Ord [21] and Moran scatterplot proposed by Anselin [13], have become systematic tools for local autocorrelation analysis. However, even the wisest are not always free from error. The Anselin's outstanding paper contains some important issues that need to be addressed. The main problems are as follows. First, *there is an unintentional mistake of mathematical reasoning resulted from step skip of mathematical transformation.* This mistake leads readers to misunderstand the relationship between global normalized spatial weight matrix and row-normalized spatial weight matrix. Second, *the row-normalized spatial weight matrix violates the distance axiom.* A spatial weight matrix is based on distance matrix or generalized distance matrix, which must conforms to distance axiom. Otherwise, the calculation result of the global or local Moran's index may appear abnormal. Third, *the basic difference between Moran's index and Geary's coefficient was omitted.* Moran's index is based on spatial population, while Geary's coefficient is based on spatial sample. Different definitions lead to different application directions. However, in the definitions of LISA, the local Geary's coefficient is based on spatial population rather spatial sample. This is not consistent with original aim of defining Geary's coefficient.

The above issues cause a series of consequences. First, *the two sets of LISA values are not equivalent to each other.* For example, the ratios of the LISA values based on non-normalized spatial weight matrix to the LISA values based on normalized spatial weight matrix are not constants. This is a serious logical problem. As we know, if two measures are equivalent to one another, the ratio of the two measures is a constant. For example, the ratio of Student's *t* statistic to Pearson's part correlation coefficient is a constant, which equals the square root of the ratio of residuals mean square deviation to total sum of squares. Second, *sometimes, the calculated values of Moran's index and Geary's coefficient exceed reasonable upper and lower limits.* Moran's index bear two sets of boundary values at least. One is absolute boundary values, that is -1 and 1, which depend on the mathematical structure of Moran's index formula and can be proved by conditional extremum principle of quadratic form. The other is relative boundary values, which are determined by the maximum and minimum eigenvalues of normalized spatial weight matrix [24–26]. Beyond the boundary values of spatial statistics is another logical problem. One of the key reasons lies in that symmetric spatial contiguity matrix is replaced by asymmetric row normalized spatial weight matrix in the process of mathematical deduction. What is more, Anselin's LISA lack clear boundary value and critical value. Anyway, spatial statistics represent a kind of measures, which may be used to describe or infer. No matter where

the goal is, a good measure should have a clear critical value or boundary value. For example, the boundary values of Pearson correlation coefficient is -1 and 1, and the critical value is 0. The purpose of this paper is to develop the spatial measures based on LISA. The rest parts are organized as below. In Section 2, Anselin's mathematical reasoning process is sorted out and his unintentional mistakes are corrected. Based on the mathematical derivation, the local Moran index and local Geary coefficient will be normalized. In addition, the strict mathematical relationship between Moran's indexes and Geary's coefficients are derived. In Section 3, the observational data of the system of cities in Beijing-Tianjin-Hebei region in China will be employed to testify the improved results. In Sections 4 and 5, the related questions are discussed, and finally, the discussion will be concluded by summarizing the main points of this study.

## 2 Theoretical results

### 2.1 Local spatial autocorrelation measurements

**2.1.1 The first formula of local Moran index.** One of the bases of spatial analysis is spatial proximity matrix, which can be measured by spatial distance matrix. Spatial distance matrix or spatial proximity matrix can be transformed into spatial contiguity matrix by means of spatial weight function such as negative power law or step function [27, 28]. A spatial contiguity matrix can be treated as non-normalized spatial weight matrix. Suppose that there are $n$ elements in a geographical region, and this size of the $i$th element is measured by $x_i$ ($i = 1, 2,\ldots, n$). The size variable $x$ are not standardized and the spatial contiguity matrix $\mathbf{V} = [v_{ij}]$ is not transformed into the globally normalized spatial weight matrix $\mathbf{W} = [w_{ij}]$. Note that the so-called global normalization refers to the normalization of a matrix or vector by the sum of its elements. So, global normalization can also be termed sum-normalization or sum-based normalization. Correspondingly, row-normalization is a type of local normalization which can also be called row-based normalization. Using the symbol systems defined in this context, we can extract two sets of local spatial autocorrelation statistics (Table 1). The first local Moran index formula defined by Anselin [19] is as follows

$$I_i^* = (x_i - \bar{x})\sum_{j=1}^{n} v_{ij}(x_j - \bar{x}) = y_i\sum_{j=1}^{n} v_{ij}y_j, \tag{1}$$

where $y_i = x_i - \bar{x}, y_j = x_j - \bar{x}$ denote centralized size variables, and $\bar{x}$ refers to mean value. In Eq (1), $i \neq j$, otherwise $v_{ij} = 0$. The centralized variables can be transformed into standardized variables by means of $z$-score formula. Based on population standard derivation, the

**Table 1. Three sets of LISAs researched in this paper based on Anselin's work.**

| Item | Index | Weight matrix | Size variable | Symbol |
|---|---|---|---|---|
| **First set of local LISA** | Local Moran's $I$ | No normalization | Centralization | MI1 |
| | Local Geary's $C$ | No normalization | Centralization | GC1 |
| **Second set of local LISA** | Local Moran's $I$ | Row normalization | Standardization based on population standard deviation | MI2 |
| | Local Geary's $C$ | Row normalization | Standardization based on population standard deviation | GC2 |
| **Third set of local LISA** | Local Moran's $I$ | Global normalization | Standardization based on population standard deviation | MI3 |
| | Local Geary's $C$ | Global normalization | Standardization based on sample standard deviation | GC3 |

**Note**: If a spatial dataset is large enough, the distinction between population standard derivation and sample standard derivation can be ignored. However, sometimes the spatial data set is not so large, and this difference cannot be ignored, otherwise biased calculation results may lead to inappropriate conclusions.

standardized variables can be expressed as

$$z_i = \frac{y_i}{\sigma} = \frac{x_i - \bar{x}}{\sigma}, \; z_j = \frac{y_j}{\sigma} = \frac{x_j - \bar{x}}{\sigma},$$

where $z$ denotes standardized variable, and $\sigma$ refers to population standard deviation. The sum of Eq (1) is

$$\sum_{i=1}^{n} I_i^* = \sum_{i=1}^{n} y_i \sum_{j=1}^{n} v_{ij} y_j = \sum_{i=1}^{n} \sum_{j=1}^{n} v_{ij} y_i y_j, \tag{2}$$

which is essentially the sum of spatially weighted outer products of centralized variables. The spatial weight coefficient is not normalized by sum. The sum of the elements in spatial contiguity matrix is

$$V_0 = \sum_{i=1}^{n} \sum_{j=1}^{n} v_{ij}. \tag{3}$$

Dividing Eq (2) by $V_0$ yields spatial precision weighted auto-covariance as follows

$$Cov = \frac{1}{\sum_{i=1}^{n} \sum_{j=1}^{n} v_{ij}} \sum_{i=1}^{n} I_i^* = \frac{1}{V_0} \sum_{i=1}^{n} \sum_{j=1}^{n} v_{ij} y_i y_j. \tag{4}$$

Furthermore, the spatial weighted covariance can be divided by the population variance of the size variable, which is called the second moment in literature [19], that is

$$\sigma^2 = \frac{1}{n} \sum_{i=1}^{n} (x_i - \bar{x})^2 = \frac{1}{n} \sum_{i=1}^{n} y_i^2. \tag{5}$$

The result is global Moran's index, $I = Cov/\sigma^2$. It can be expanded as

$$I = \frac{\frac{1}{\sum_{i=1}^{n} \sum_{j=1}^{n} v_{ij}} \sum_{i=1}^{n} I_i^*}{\frac{1}{n} \sum_{i=1}^{n} y_i^2} = \frac{n \sum_{i=1}^{n} \sum_{j=1}^{n} v_{ij} y_i y_j}{V_0 \sum_{i=1}^{n} y_i^2} = \frac{1}{\sigma^2 V_0} \sum_{i=1}^{n} \sum_{j=1}^{n} v_{ij} y_i y_j = \sum_{i=1}^{n} \sum_{j=1}^{n} w_{ij} z_i z_j, \tag{6}$$

where $w_{ij}$ is the element of the globally normalized weight matrix $\mathbf{W}$. According to Anselin [19], Eq (6) can be expressed as

$$I = \frac{1}{\sigma^2 V_0} \sum_{i=1}^{n} I_i^*. \tag{7}$$

The relationship between the sum of Anselin's first local Moran's indexes and the global Moran's index is obtained as below

$$\sum_{i=1}^{n} I_i^* = \sigma^2 V_0 I = \gamma I. \tag{8}$$

The proportionality coefficient in Eq (8) is

$$\gamma = \sigma^2 V_0 = \left(\frac{1}{n}\sum_{i=1}^{n} y_i^2\right)\left(\sum_{i=1}^{n}\sum_{j=1}^{n} v_{ij}\right), \tag{9}$$

which represents the general expression of the ratio of the sum of local Moran's indexes to the global Moran's index. Please note that Eqs (8) and (9) are derived from the relations based on non-normalized spatial weight matrix. They cannot be directly applied to the mathematical processes based on row-normalized spatial weight matrix. According to Anselin [19], Eq (3) can be replaced by a vector indicating the sum of rows of the spatial contiguity matrix as below

$$V_i = \sum_{j=1}^{n} v_{ij}. \tag{10}$$

Correspondingly, spatial contiguity matrix can be normalized by row. Anselin called it row-standardized spatial weights matrix [19]. In this way, Eq (4) becomes a locally weighted spatial auto-covariance, that is

$$Cov_i = \frac{I_i^*}{\sum_{j=1}^{n} v_{ij}} = \frac{1}{V_i}\sum_{j=1}^{n} v_{ij} y_i y_j. \tag{11}$$

The summation of Eq (11) is

$$\sum_{i=1}^{n} Cov_i = \sum_{i=1}^{n} \frac{I_i^*}{V_i} = \sum_{i=1}^{n}\sum_{j=1}^{n} \frac{v_{ij}}{V_i} y_i y_j. \tag{12}$$

Based on Eqs (11) and (12), it is impossible to obtain the global spatial weighted auto-covariance, and it is impossible to derive the simple summation relationship between local Moran index and global Moran index. If so, the reasoning from Eq (4) to Eq (9) will be invalid.

It can be seen that the local-global relationship based on Anselin's first local Moran index formula suggests a global normalized weight matrix with symmetry. The first local Moran index formula of Anselin [19] is correct, it satisfy the two requirements defined by Anselin [19]. The shortcoming lies in that it is not standardized. A good measure should have a clear critical value (reference value) or a pair of explicit boundary values. However, the local Moran index calculated by Eq (1) has neither boundary values nor clear threshold value.

**2.1.2 The second formula of local Moran index.** Suppose that the variables are standardized, the spatial contiguity matrix is transformed into a spatial weight matrix which is normalized by row. In this way, $V_0$ in is replaced by $V_i$ in Eq (4). Thus, revised Eq (4) divided by population variance yields the second local Moran's index formula of Anselin [19], $I_i^{**} = Cov_i/\sigma^2$, that is

$$I_i^{**} = \frac{1}{\sigma^2}\sum_{j=1}^{n} \frac{v_{ij}}{V_i} y_i y_j = \frac{1}{\sigma^2} y_i \sum_{j=1}^{n} w_{ij}^* y_j, \tag{13}$$

where $w_{ij}^*$ denotes the elements in the row-normalized spatial weight matrix, $\mathbf{V}^*$. Apparently, Eq (13) is based on Eqs (10) and (11). Thus, in terms of Eq (10), the sum of the spatial weight

matter is

$$V_0^* = \sum_{i=1}^{n} \sum_{j=1}^{n} \frac{v_{ij}}{V_i} = \sum_{i=1}^{n} \left( \frac{1}{V_i} \sum_{j=1}^{n} v_{ij} \right) = \sum_{i=1}^{n} (1) = n. \tag{14}$$

The variance of standardized variable is 1, namely, $\sigma^2 = 1$. For normalized matrix by row, the sum is $V_0^* = n$, thus we have

$$\gamma = \sigma^2 V_0^* = V_0^* = n. \tag{15}$$

Substituting Eq (15) into Eq (8) seems to yield the following relation

$$\sum_{i=1}^{n} I_i^{**} = nI, \tag{16}$$

which is once of relations given by Anselin [19]. Note that the symbols have been slightly changed. That is, $V_0$ is replaced by $V_0^*$, and $I_i^*$ is replaced by $I_i^{**}$. The new added asterisk indicates the inherent difference between the two sets of local Moran's indexes. On the surface, there is no problem at all in the mathematical derivation process. However, Anselin [19] inadvertently made a mistake in above reasoning process (S1 File). Looking at Eq (14) alone, we may think that there is no problem. However, by summing Eq (13), it is impossible to extract an independent Eq (14), and this is exactly the problem. In fact, Anselin [19] unintentionally replaced a mathematical concept by directly applying the derived results based on non-normalized weight matrices to the relationship formula based on row-normalized spatial weight matrices. Regardless of whether the spatial contiguity matrix is symmetric or not, the non-normalized spatial weight matrix and the row normalized spatial weight matrix are not isomorphic to each other. However, the non-normalized spatial weight matrix is isomorphic to the sum-based normalized spatial weight matrix.

Mathematical deduction problems can be revealed through logical analysis, and also can be reflected through empirical analysis. Let us check the problem from another view of angle. The relation between the second set of local Moran's indexes of Anselin [19] and global Moran's index can be derived from Eq (13). The summation of the local Moran's indexes based on Eq (13) is

$$\sum_{i=1}^{n} I_i^{**} = \frac{1}{\sigma^2} \sum_{i=1}^{n} \sum_{j=1}^{n} \frac{v_{ij}}{V_i} y_i y_j = V_0 \sum_{i=1}^{n} \sum_{j=1}^{n} \frac{w_{ij}}{V_i} z_i z_j = \sum_{i=1}^{n} \sum_{j=1}^{n} w_{ij}^* z_i z_j. \tag{17}$$

By variable standardization, the population standard deviation becomes 1 unit, i.e., $\sigma^2 = 1$. However, the row sum of spatial contiguity matrix $V_i$ is not a constant. It can neither be eliminated nor converted to a constant. Therefore, no constant proportionality relation between the second set of local Moran's index and the global Moran's index. If and only if Eq (6) is introduced into Eq (17) can the proportional relationship similar to Eq (8) be derived. Based on Eq (6), Eq (17) can be re-expressed as

$$\sum_{i=1}^{n} I_i^{**} = \frac{\sum_{i=1}^{n} \sum_{j=1}^{n} w_{ij}^* z_i z_j}{\sum_{i=1}^{n} \sum_{j=1}^{n} w_{ij} z_i z_j} I. \tag{18}$$

Unfortunately, we cannot prove the following relation:

$$\sum_{i=1}^{n}\sum_{j=1}^{n}w_{ij}^{*}z_{i}z_{j} = n\sum_{i=1}^{n}\sum_{j=1}^{n}w_{ij}z_{i}z_{j} = nI. \tag{19}$$

This lends further support to the judgment that Eq (16) does not hold. However, the proportional relationship given in Eqs (17) and (18) can be easily verified by the observational data. Another view of angle is to examine the ratios of two sets of local Moran indices. If the ratios are constant, the two definitions are equivalent to one another, otherwise they are not. In fact, the values in the first set of local Moran indexes divided by the corresponding values in the second set of local Moran indexes yields

$$\frac{I_{i}^{*}}{I_{i}^{**}} = \frac{\sigma^{2}\sum_{j=1}^{n}v_{ij}y_{i}y_{j}}{\sum_{j=1}^{n}\frac{v_{ij}}{V_{i}}y_{i}y_{j}} = \sigma^{2}V_{i}, \tag{20}$$

which, obviously, is a variable that changes with $V_i$ rather than a constant.

It can be seen that the ratios of two sets of local Moran's indexes are not constant, so they are not equivalent to each other. This suggests that, the second set of local Moran indexes cannot satisfy the second requirement of Anselin [19], which said, "The sum of the local indicators is proportional to a global indicator". The reason for the fault is that Anselin [19] inadvertently replaced a concept in this mathematical derivation. Concretely speaking, the globally normalized symmetric weight matrix **W** becomes the locally normalized asymmetric weight matrix **V**[*]. This way violates the law of identity of concepts and the principle of logical consistency in mathematical reasoning.

**2.1.3 The formula of local Geary coefficient.** The global Geary coefficient is complementary to the global Moran index: the former is oriented to spatial sample analysis, and the latter is based on spatial statistical population. Similar to the treatment of local Moran index, two local Geary statistics were defined by Anselin [19]. It is assumed that the variables are not standardized and the spatial contiguity matrix is not transformed into a global normalized spatial weight matrix. Anselin [19] defined the first local Geary's coefficient as

$$C_{i}^{*} = \sum_{j=1}^{n}v_{ij}(y_{i} - y_{j})^{2}, \tag{21}$$

in which the divisor 2 is ignored. Suppose that the variable is standardized, and the spatial contiguity matrix is transformed into a row normalized spatial weight matrix. Anselin [19] defines the second local Geary coefficient as

$$C_{i}^{**} = \frac{1}{\sigma^{2}}\sum_{j=1}^{n}w_{ij}^{*}(y_{i} - y_{j})^{2}. \tag{22}$$

Summation of Eq (21) divided by the population variance $\sigma^2$ is

$$\frac{1}{\sigma^{2}}\sum_{i=1}^{n}C_{i}^{*} = \frac{n\sum_{i=1}^{n}\sum_{j=1}^{n}v_{ij}(y_{i} - y_{j})^{2}}{\sum_{i=1}^{n}y_{i}^{2}} = \frac{2nV_{0}}{n-1}\frac{(n-1)\sum_{i=1}^{n}\sum_{j=1}^{n}v_{ij}(y_{i} - y_{j})^{2}}{2V_{0}\sum_{i=1}^{n}y_{i}^{2}} = \gamma_{c}C, \tag{23}$$

where $C$ refers to global Geary coefficient. It can be expressed as

$$C = \frac{(n-1)\sum_{i=1}^{n}\sum_{j=1}^{n}v_{ij}(y_i - y_j)^2}{2V_0\sum_{i=1}^{n}y_i^2} = \frac{1}{2s^2}\sum_{i=1}^{n}\sum_{j=1}^{n}w_{ij}(y_i - y_j)^2 = \frac{1}{2}\sum_{i=1}^{n}\sum_{j=1}^{n}w_{ij}(z_i^* - z_j^*)^2. \quad (24)$$

in which $z^*$ referes to the standardized size variable based on the sample standard deviation $s$, i.e.,

$$z_i^* = \frac{y_i}{s} = \frac{x_i - \bar{x}}{s}, \; z_j^* = \frac{y_j}{s} = \frac{x_j - \bar{x}}{s}.$$

Here $s$ denotes sample standard deviation, that is, $s = \sigma(n/(n\text{-}1))^{1/2}$. In addition, the proportional coefficient between the sum of the first local Geary coefficient divided by the population variance and the global Geary coefficient is as below

$$\gamma_c = \frac{2nV_0}{n-1}. \quad (25)$$

Therefore, the relationship between the sum of the first local Geary coefficients and the global Geary coefficients is

$$\sum_{i=1}^{n}C_i^* = \frac{2nV_0\sigma^2}{n-1}C = \gamma_c\sigma^2 C. \quad (26)$$

This formula is correct, and it satisfies the two requirements given by Anselin [19]. However, it is neither direct nor standard. Dividing the summation of Eq (21) by both the population variance $\sigma^2$ and the sum of the spatial weight matrix $V_0$ to obtain the relationship between the local Geary's coefficients and the global Geary coefficient, that is

$$\sum_{i=1}^{n}C_i^{**} = \frac{n\sum_{i=1}^{n}\sum_{j=1}^{n}v_{ij}(y_i - y_j)^2}{V_0\sum_{i=1}^{n}y_i^2} = \frac{2n}{n-1}\frac{(n-1)\sum_{i=1}^{n}\sum_{j=1}^{n}w_{ij}(y_i - y_j)^2}{2\sum_{i=1}^{n}y_i^2} = \frac{2n}{n-1}C. \quad (27)$$

This is the corrected expression of the relationship between local Geary coefficient and global Geary's coefficient, differing from that given by Anselin [19]. The reason is that derivation of this relationship is based on the global normalization of spatial weight matrix. However, due to the fact that divisor 2 is ignored in Eq (21), when $n$ is sufficiently large in Eq (27), the sum of local Geary's coefficients does not equal the global Geary's coefficient. Based on the row-normalized weight matrix, the sum of local Geary's coefficients is

$$\sum_{i=1}^{n}C_i^{**} = \frac{n\sum_{i=1}^{n}\frac{1}{V_i}\sum_{j=1}^{n}v_{ij}(y_i - y_j)^2}{\sum_{i=1}^{n}y_i^2} = \frac{nV_0\sum_{i=1}^{n}\frac{1}{V_i}\sum_{j=1}^{n}w_{ij}(y_i - y_j)^2}{\sum_{i=1}^{n}y_i^2}. \quad (28)$$

The constant proportional relationship between local Geary coefficient and global Geary coefficient cannot be derived in terms of Eq (28). Anselin [19] believes that, according to Eq (25), for the weight matrix normalized by row, $V_0 = n$, so there is $\gamma_c = 2n^2/(n\text{-}1)$, that's right.

Then he gave the following relation

$$\sum_{i=1}^{n} C_i^{**} = \frac{2n^2}{n-1} C = \gamma_c C. \tag{29}$$

This is wrong and cannot be strictly derived by mathematical methods, nor can it be verified by observational data. Based on the row-normalized weight matrix, the correct result is

$$\sum_{i=1}^{n} C_i^{**} = \frac{2n}{n-1} \frac{\sum_{i=1}^{n}\sum_{j=1}^{n} w_{ij}^* (z_i^* - z_j^*)^2}{\sum_{i=1}^{n}\sum_{j=1}^{n} w_{ij} (z_i^* - z_j^*)^2} C = \gamma_c^* C, \tag{30}$$

in which $\gamma_c^*$ represents the proportionality coefficient. The coefficient can be expressed as

$$\gamma_c^* = \frac{2n}{n-1} \frac{\sum_{i=1}^{n}\sum_{j=1}^{n} w_{ij}^* (z_i^* - z_j^*)^2}{\sum_{i=1}^{n}\sum_{j=1}^{n} w_{ij} (z_i^* - z_j^*)^2}, \tag{31}$$

which is not a constant. It cannot be proved that Eq (29) is equivalent to Eq (30). Moreover, starting from Eqs (21) and (22), the proportional relationship between the two sets of local Geary coefficients is

$$\frac{C_i^*}{C_i^{**}} = \frac{\sigma^2 \sum_{j=1}^{n} v_{ij} (y_i - y_j)^2}{\sum_{j=1}^{n} \frac{v_{ij}}{V_i} (y_i - y_j)^2} = \sigma^2 V_i = \frac{I_i^*}{I_i^{**}}. \tag{32}$$

This is obviously not a constant, but a variable that changes with the sum of the rows of the spatial proximity matrix. This shows that the two sets of local Geary coefficients are not equivalent to each other, and the ratio of the corresponding values of the two sets of local Geary coefficients is equal to the ratio of the values of the two sets of local Moran's indices. In short, the second set of local Geary statistic does not satisfy the second requirement given by Anselin [19].

## 2.2 Revised and normalized results

### 2.2.1 Adjustment of symbol system and clarification of concept.
Concept is the cornerstone of logic. If and only if a concept is clear, there will be no mistakes in reasoning. The premise of mathematical reasoning is the symbolization of concepts. Confusion of symbols can easily lead to mistakes in reasoning. The main reason for the inconsistency between the two sets of LISA proposed by Anselin [19] is the unintentional concept substitution caused by the symbol mixing of spatial measure matrixes. At present, there are several problems about spatial autocorrelation in geographical literature.

Firstly, the symbols of the spatial weight matrix need to be improved. The symbols of spatial contiguity matrix (SCM), say, $[1/d_{ij}]$, and those of spatial weight matrix (SWM), say, $[v_{ij}/\Sigma\Sigma v_{ij}]$, where $v_{ij} = 1/d_{ij}$, are confused with each other. The two matrixes are regarded as equivalence and are both represented by the same symbol $[w_{ij}]$. In fact, the spatial distance matrix can be transformed into a spatial contiguity matrix according to a certain distance decay function, and the weight matrix can be obtained by normalizing the spatial contiguity matrix [29].

Despite the final result is the same in the case of symbol confusion, the expression form causes many unnecessary misunderstandings for beginners. This paper distinguishes the symbols as follows: SCM is represented by $\mathbf{V}$, its elements are represented by $v_{ij}$; SWM is represented by $\mathbf{W}$, and its elements are expressed as $w_{ij}$. Thus we have SCM, $\mathbf{V} = [v_{ij}]$, and SWM, $\mathbf{W} = [w_{ij}] = [v_{ij}/\Sigma\Sigma v_{ij}]$.

Secondly, the definitions of spatial matrixes need to be explained. After the spatial contiguity matrix (SCM) is transformed into the spatial weight matrix (SWM), the global normalization and local normalization by row are confused. Anselin [19], the original founder of the local Moran index, adopted the method of row normalization (he term the processing "row-standardization"). The sum of the SWM elements is thus equal to $n$. However, this method will lead to two results: (1) The symmetry of the spatial distance matrix is broken. Spatial weight matrix comes from spatial distance matrix or generalized spatial distance matrix. One of the important properties of distance measure is symmetry: $d_{ij} = d_{ji}$ holds for all $i$ and $j$ [30]. This is one of the four principles of the distance axioms (positivity, specification, symmetry, and triangle inequality). (2) The absolute value of the calculated local Moran index may exceed 1 sometimes. Moran index is an autocorrelation coefficient whose absolute value should fall between —1 and 1 in theory. As for the special boundary values of Moran's index determined by the maximum and minimum eigenvalues of the spatial weight matrix, it should be discussed in another work.

Thirdly, the meanings and symbols of the two types of variance are different. The population variance is often confused with the sample variance in spatial statistics. Moran's index is defined based on population variance, and Geary's coefficient is defined based on sample variance [29]. According to Fisher's symbol system in statistics, the population variance is expressed as $\sigma^2$, and the denominator in the formula is $n$; the sample variance is expressed as $s^2$, and the denominator in the formula is $n$-1 in the formula [31]. The relationship between them is $\sigma^2 = (n\text{-}1)s^2/n$.

Fourth, the difference in numbering between rows and columns needs to be noted. There is sometimes confusion between row summation and column summation. The sum based on row vector is expressed as summation by $j$, and the sum of column vector is expressed as summation by $i$. Based on globally normalized weight matrix, the difference is only formal and has nothing to do with the results. However, based on row-normalized weight matrix, the results of row summation differs from the results of column summation.

Fifth, the methods of value transformation need to be particularly clarified. The concepts of normalization and standardization are always confused in literature. Generalized standardization includes normalization. However, both standardization and normalization have different definition methods and corresponding calculation formulas. The transformation formula of variables should be determined according to different research objectives (S2 File).

In order to make it easy for readers to understand, it is necessary to distinguish symbols, and then clarify the concept of variable transformation. There are three principles for adopting symbols in this paper: First, the principle of consensus. Priority will be given to the conventional expression in the field of mathematical statistics. For example, the population standard deviation is expressed as $\sigma$, and the sample standard deviation is expressed as $s$ [31]. Second, the principle of direction. For example, the spatial weight matrix represents $\mathbf{W}$ because "W" it is the capital form of the initial of "weight". Third, the principle of distinction. For example, the spatial contiguity matrix represents $\mathbf{V}$, so as to distinguish it from the spatial weight matrix $\mathbf{W}$, and this distinguishing facilitates mathematical reasoning. Among the above three principles, the distinction principle is the most important (Table 2). In the spatial autocorrelation literature, centralization variables (such as defining local Moran's index), standardized variables (such as simplifying the calculation of global Moran index) and globally normalized variables

**Table 2. Comparison between Anselin's symbol system and the symbol system in this paper.**

| Measure set | Anselin | This paper |
|---|---|---|
| **Spatial proximity matrix (SPM)** | – | $\mathbf{U} = \{d_{ij}\}$ |
| **Spatial contiguity matrix (SCM): non-normalized SWM** | $W = \{w_{ij}\}$ | $\mathbf{V} = \{v_{ij}\}$ |
| **Row-normalized spatial weight matrix (RSWM)** | $W = \{w_{ij}\}$ | – |
| **Sum-normalized spatial weight matrix (SSWM)** | – | $\mathbf{W} = \{w_{ij}\}$ |
| **Row-normalized spatial weight matrix** | $W = \{w_{ij}\}$ | – |
| **Sum of elements of spatial contiguity matrix** | $S_0$ | $V_0$ |
| **Sum of elements of spatial weight matrix** | $S_0$ | $W_0$ |
| **Size variable** | – | $x_i, x_j$ |
| **Centralized variable** | $z_i, z_j$ | $y_i, y_j$ |
| **Standardized variable** | – | $z_i, z_j$ |
| **Population variance** | $m_2$ | $\sigma^2$ |
| **Sample variance** | – | $s^2$ |
| Global Moran's $I$ | $I$ | $I$ |
| Local Moran's $I$ | $I_i$ | $I_i$ |
| Global Geary's $I$ | $c$ | $C$ |
| Local Geary's $I$ | $c_i$ | $C_i$ |

**Note**: In the context, the sum-normalized spatial weight matrix is also termed sum-based normalized spatial weight matrix or globally normalized spatial weight matrix by sum. Correspondingly, the row-normalized spatial weight matrix is also called row-based normalized spatial weight matrix or locally normalized spatial weight matrix by row.

(such as simplifying the calculation of Getis-Ord's index) are used, respectively (Table 3). In the literature, when the spatial weight matrix is normalized by row, the concept of row standardization is adopted, but the calculation formula is not given [19]. This can easily lead to misunderstandings for beginners of spatial autocorrelation analysis.

**2.2.2 Definition of normalized local Moran's index.** Moran's index is defined on the basis of population standard deviation rather than sample standard deviation. Accordingly, local Moran's index should also be defined through population standard deviation. In light of Eq (7), canonical local Moran's index can be defined as

$$I_i = \frac{I_i^*}{\sigma^2 V_0} = \frac{1}{\sigma^2} y_i \sum_{j=1}^{n} \frac{v_{ij}}{V_0} y_j = z_i \sum_{j=1}^{n} w_{ij} z_j. \tag{33}$$

Further, according to Eq (7), the relation between global Moran's index and the sum of local Moran's indexes is

$$I = \sum_{i=1}^{n} \left( \frac{I_i^*}{\sigma^2 V_0} \right) = \sum_{i=1}^{n} I_i. \tag{34}$$

**Table 3. Value transformation methods, calculation formulas, and properties of converted variables.**

| Method | Calculation formula | Property |
|---|---|---|
| **Centralization** | $y_i = x_i - \bar{x}$ | The mean value is 0 |
| Standardization by $z$-score | $z_i = (x_i - \bar{x})/\sigma$, $z_i^* = (x_i - \bar{x})/s$, | The mean value is 0 and the standard deviation is 1 |
| **Range normalization** | $x_i^{(r)} = (x_i - x_{min})/(x_{max} - x_{min})$ | The values range from 0 to 1 |
| **Global normalization** | $x_i^{(t)} = x_i/\sum_i x_i$, $w_{ij} = v_{ij}/\sum_i \sum_j v_{ij}$ | The values come between 0 and 1 and the sum of the values equals 1 |

According to Eq ([33]), the relation between Anselin's first set of local Moran indexes and the local Moran's indexes formula improved in this paper is

$$I_i^* = \gamma I_i = \sigma^2 V_0 I_i. \tag{35}$$

Thus, for the globally normalized spatial weight matrix $\mathbf{W}$ and the standardized variable based on population standard deviation $\mathbf{z}$, we have $\sigma^2 = 1$, $V_0 = 1$. Thus, Eq ([9]) should be replaced by

$$\gamma_0 = \sigma^2 V_0 = \left(\frac{1}{n} \sum_{i=1}^n z_i^2\right)\left(\sum_{i=1}^n \sum_{j=1}^n w_{ij}\right) = 1. \tag{36}$$

This suggests that, according to the second basic requirement for LISA from Anselin [19], the sum of normalized local Moran's index equals the global Moran's index.

**2.2.3 Definition of normalized local Geary's coefficient.** Geary's coefficient is defined on the basis of sample standard deviation rather than population standard deviation. Accordingly, local Geary's coefficient should also be defined through sample standard deviation. The generalized Geary's coefficient is another case [29]. In terms of Eq ([26]), global Geary's coefficient can be expressed as

$$C = \frac{n-1}{2n V_0 \sigma^2} \sum_{i=1}^n C_i^* = \frac{1}{2 V_0 s^2} \sum_{i=1}^n C_i^* = \sum_{i=1}^n \left(\frac{C_i^*}{2 V_0 s^2}\right) = \sum_{i=1}^n C_i, \tag{37}$$

where $s^2 = n\sigma^2/(n\text{-}1)$ reflects the relationship between sample variance $s^2$ and population variance $\sigma^2$. Thus local Geary's coefficient can be defined as

$$C_i = \frac{C_i^*}{2 V_0 s^2} = \frac{1}{2 V_0 s^2} \sum_{j=1}^n v_{ij}(y_i - y_j)^2 = \frac{1}{2} \sum_{j=1}^n w_{ij}(z_i^* - z_j^*)^2. \tag{38}$$

Summing Eq ([38]) yields global Geary's coefficient, that is, Eq ([24]). According to Eq ([37]), the relation between Anselin's first set of Geary's coefficient and the local Geary's coefficient formula improved in this paper is

$$C_i^* = \gamma_c \sigma^2 C_i = 2 s^2 V_0 C_i. \tag{39}$$

Thus, for the globally normalized spatial weight matrix $\mathbf{W}$ and the standardized vector based on sample standard deviation $\mathbf{z}^*$, we have $s^2 = 1$, $V_0 = 1$. Thus, according to Eq ([26]), the relation between proportionality coefficients is

$$\gamma_c \sigma^2 = 2 s^2 V_0 = 2. \tag{40}$$

Moran's index and Geary's coefficient reflect the same problem from different angles of view. It can be proved that the relationship between global Moran's $I$ and global Geary's $C$ is as follows

$$C = \frac{\sum_{i=1}^n \sum_{j=1}^n v_{ij} y_i^2 - \sum_{i=1}^n \sum_{j=1}^n v_{ij} y_i y_j}{V_0 \frac{1}{n-1} \sum_{i=1}^n y_i^2} = \frac{n-1}{n}(\mathbf{o}^{\mathrm{T}} \mathbf{W} \mathbf{z}^2 - \mathbf{z}^{\mathrm{T}} \mathbf{W} z) = \frac{n-1}{n}(\mathbf{o}^{\mathrm{T}} \mathbf{W} \mathbf{z}^2 - I), \tag{41}$$

where $\mathbf{z}$ denotes standardized vector based on population standard deviation, $\mathbf{z}^2 = \mathrm{diag}(\mathbf{z}\mathbf{z}^{\mathbf{T}})$ refers to a vector composed of the squares of the elements in $\mathbf{z}$, $\mathbf{o}^{\mathrm{T}} = [1\ 1\ \dots\ 1]$ is a ones vector

in which all the elements are 1. The symbol "T" indicates transposition, and the function "diag" represents taking the diagonal elements of a matrix to form a vector. If the mean of the global Moran's index is treated as $I_0 = 1/(1\text{-}n)$, the mean of global Geary's coefficient, $C_0$, can be estimated by

$$C_0 = \frac{n-1}{n}(\mathbf{e}^{\mathrm{T}}\mathbf{W}\mathbf{z}^2 - I_0) = \frac{n-1}{n}(\mathbf{e}^{\mathrm{T}}\mathbf{W}\mathbf{z}^2 - \frac{1}{1-n}) = \frac{n-1}{n}\mathbf{e}^{\mathrm{T}}\mathbf{W}\mathbf{z}^2 + \frac{1}{n}. \qquad (42)$$

Further, the relationship between local Moran's indexes and local Geary's coefficient can be derived. From Eq (38) it follows

$$C_i = \frac{1}{2V_0\frac{n}{n-1}\sigma^2}\sum_{j=1}^{n} v_{ij}(y_i - y_j)^2 = \frac{n-1}{2n}\sum_{j=1}^{n} w_{ij}(z_i - z_j)^2. \qquad (43)$$

Changing the form of Eq (43) yields

$$C_i = \frac{n-1}{2n}(\sum_{j=1}^{n} w_{ij}(z_i^2 + z_j^2) - 2\sum_{j=1}^{n} w_{ij}z_iz_j) = \frac{n-1}{2n}(\sum_{j=1}^{n} w_{ij}(z_i^2 + z_j^2) - 2I_i). \qquad (44)$$

This means that there is a strict numerical conversion relationship between local Moran's indexes and local Geary's coefficient, although they describe the same problem from different angles. It can be seen that Eq (41) can be obtained by summing Eq (44).

In the new framework for LISA, the spatial weight matrix is normalized by sum. This is a type of global normalization in value transformation. There are several benefits to using a globally normalized weight matrix. We know that mathematics is a science relying highly on form in a sense. The same mathematical method often has vastly different effects when expressed in different forms. For spatial autocorrelation, using a normalized spatial weight matrix instead of a non-normalized weight matrix results in at least the following advantages. First, by normalized weight matrix, it is very convenient to calculate the global Moran's index $I$ and local Moran's indexes $I_i$, and reflect the clear relationship between the two, $I$ and $I_i$ [29]. Second, normalizing weight matrix, we can obtain a standardized Moran's scatterplot, where the slope of the trend line is exactly equal to the global Moran's index value [32]. Third, based on normalized weight matrix, the structure of the parameters of the spatial autoregressive models can be clearly revealed using the spatial autocorrelation coefficients. Fourth, it makes the values of local Moran's index and local Geary's coefficient more intuitive. The fourth advantage mentioned above is more relevant to the research in this work. Many basic measures and models of spatial statistical analysis are rooted in conventional statistics and are created by analogy with time series analysis methods. The common measures and models of time series analysis, such as autocorrelation coefficients and autoregressive models, are also rooted in traditional statistical theories. The development of statistics took place in the wider context of the Victorian culture of measurement [31]. For simplicity's sake, the numerous data of measurement results are usually condensed into an index [33]. In this case, an index is often treated as a characteristic measurement [6, 34]. A good index either has a pair of clear boundary values, a clear critical value, or even a combination of both. Based on standardized variable and globally normalized spatial weight matrix, the values of the local Moran's indexes fall between -1 and 1, the corresponding critical value is 0; and the values of the local Geary's coefficient falls between 0 and 2, and the corresponding critical value is 1.

## 3 Empirical analysis

### 3.1 Study area and data

The results of mathematical deduction ultimately need to be verified through mathematical reasoning and empirical analysis. After all, the success of sciences rests with their great emphasis on the role of quantifiable data and their interplay with models [35]. Taking cities in Beijing, Tianjin and Hebei (BTH) region as an example, we can make a concise calculation case study. This is a demonstrative case, not an explanatory case. In other words, this example is used to verify the reasoning results rather than to study the spatial structure and characteristics of BTH urban systems. The study area includes Beijing city, Tianjin city, and the main cities of Hebei Province. The study region is also termed Jing-Jin-Ji (JJJ) region in literature [36]. The cities are all of prefecture level and above, and the number of cities is $n = 13$. The size measurement is the city population of the fifth census in 2000 and the sixth census in 2010. Town population is not taken into account. At present, urban population has the definitions of regional total population, municipal population, city population and urban population consisting city population and town population. This case uses the city population, which can better reflect the characteristics of city size. City population size can be reflected by night light area in map [32, 36]. The population size was processed by centralization ($y$), population-based standardization ($z$) and sample-based standardization ($z^*$) (Table 4). As for the spatial weight matrix, the basic data is derived from the traffic mileage between cities (Table 5). The spatial weight function adopts the special negative power law, the inverse proportion function, which is actually the intersection of power law and hyperbolic function. Thus, the spatial contiguity is defined as

$$v_{ij} = \begin{cases} 1/d_{ij}, & i \neq j \\ 0, & i = j \end{cases}, \tag{45}$$

where $d_{ij}$ denotes the distance by road between city $i$ and city $j$. On this basis, the traffic

**Table 4. Beijing-Tianjin-Hebei city population and its centralization and standardization results.**

| City | 2000 | | | | 2010 | | | |
|---|---|---|---|---|---|---|---|---|
| | $x$ | $y$ | $z$ | $z^*$ | $x$ | $y$ | $z$ | $z^*$ |
| Beijing | 949.6688 | 769.1377 | 2.9976 | 2.8800 | 1555.2378 | 1284.2528 | 2.9870 | 2.8698 |
| Tianjin | 531.3702 | 350.8391 | 1.3673 | 1.3137 | 885.6234 | 614.6384 | 1.4296 | 1.3735 |
| Shijiazhuang | 193.0579 | 12.5268 | 0.0488 | 0.0469 | 275.6871 | 4.7021 | 0.0109 | 0.0105 |
| Tanshan | 140.3887 | -40.1424 | -0.1564 | -0.1503 | 163.7579 | -107.2271 | -0.2494 | -0.2396 |
| Qinhuangdao | 70.7267 | -109.8044 | -0.4279 | -0.4112 | 95.1872 | -175.7978 | -0.4089 | -0.3928 |
| Handan | 107.1068 | -73.4243 | -0.2862 | -0.2749 | 111.7417 | -159.2433 | -0.3704 | -0.3558 |
| Xingtai | 53.6282 | -126.9029 | -0.4946 | -0.4752 | 63.7797 | -207.2053 | -0.4819 | -0.4630 |
| Baoding | 90.2496 | -90.2815 | -0.3519 | -0.3381 | 98.0177 | -172.9673 | -0.4023 | -0.3865 |
| Zhangjiakou | 79.6580 | -100.8731 | -0.3931 | -0.3777 | 90.0218 | -180.9632 | -0.4209 | -0.4044 |
| Chengde | 32.5821 | -147.9490 | -0.5766 | -0.5540 | 49.8293 | -221.1557 | -0.5144 | -0.4942 |
| Cangzhou | 44.3561 | -136.1750 | -0.5307 | -0.5099 | 48.9701 | -222.0149 | -0.5164 | -0.4961 |
| Langfang | 29.5879 | -150.9432 | -0.5883 | -0.5652 | 46.6539 | -224.3311 | -0.5218 | -0.5013 |
| Hengshui | 24.5229 | -156.0082 | -0.6080 | -0.5842 | 38.2976 | -232.6874 | -0.5412 | -0.5200 |
| **Mean** | **180.5311** | **0.0000** | **0.0000** | **0.0000** | **270.9850** | **0.0000** | **0.0000** | **0.0000** |
| $\sigma$ | **256.5845** | **256.5845** | **1.0000** | **0.9608** | **429.9496** | **429.9496** | **1.0000** | **0.9608** |
| $s$ | **267.0616** | **267.0616** | **1.0408** | **1.0000** | **447.5057** | **447.5057** | **1.0408** | **1.0000** |

Table 5. Spatial distance matrix ($d_{ij}$) of Beijing-Tianjin-Hebei cities based on traffic mileage.

| City | Beijing | Tianjin | Shijiazhuang | Tanshan | Qinhuangdao | Handan | Xingtai | Baoding | Zhangjiakou | Chengde | Cangzhou | Langfang | Hengshui |
|---|---|---|---|---|---|---|---|---|---|---|---|---|---|
| Beijing | 0 | 160.8855 | 321.7625 | 185.4770 | 288.9055 | 479.9810 | 430.2520 | 187.1300 | 198.1975 | 194.5940 | 233.4440 | 83.2755 | 299.7580 |
| Tianjin | 160.8855 | 0 | 344.5825 | 101.4105 | 242.6355 | 454.8400 | 425.3890 | 201.9420 | 332.9375 | 280.6470 | 138.6135 | 86.1555 | 259.8555 |
| Shijiazhuang | 321.7625 | 344.5825 | 0 | 423.7510 | 568.1560 | 167.2815 | 114.0840 | 138.9090 | 430.8215 | 506.6400 | 221.7565 | 283.2495 | 142.5935 |
| Tanshan | 185.4770 | 101.4105 | 423.7510 | 0 | 151.3880 | 547.4205 | 517.8910 | 289.5120 | 376.8000 | 185.3500 | 215.0285 | 144.6130 | 352.4360 |
| Qinhuangdao | 288.9055 | 242.6355 | 568.1560 | 151.3880 | 0 | 711.7120 | 662.2960 | 433.9170 | 481.3360 | 222.2030 | 375.5205 | 292.9180 | 508.4835 |
| Handan | 479.9810 | 454.8400 | 167.2815 | 547.4205 | 711.7120 | 0 | 53.4600 | 296.7465 | 606.6940 | 664.8585 | 335.0465 | 440.4685 | 214.2995 |
| Xingtai | 430.2520 | 425.3890 | 114.0840 | 517.8910 | 662.2960 | 53.4600 | 0 | 245.8830 | 557.3515 | 615.1295 | 299.4430 | 391.1260 | 167.0325 |
| Baoding | 187.1300 | 201.9420 | 138.9090 | 289.5120 | 433.9170 | 296.7465 | 245.8830 | 0 | 278.0950 | 372.0075 | 150.5130 | 147.8300 | 144.8405 |
| Zhangjiakou | 198.1975 | 332.9375 | 430.8215 | 376.8000 | 481.3360 | 606.6940 | 557.3515 | 278.0950 | 0 | 372.8730 | 411.7425 | 257.5700 | 455.2955 |
| Chengde | 194.5940 | 280.6470 | 506.6400 | 185.3500 | 222.2030 | 664.8585 | 615.1295 | 372.0075 | 372.8730 | 0 | 407.1040 | 259.8085 | 495.3555 |
| Cangzhou | 233.4440 | 138.6135 | 221.7565 | 215.0285 | 375.5205 | 335.0465 | 299.4430 | 150.5130 | 411.7425 | 407.1040 | 0 | 149.7245 | 140.0620 |
| Langfang | 83.2755 | 86.1555 | 283.2495 | 144.6130 | 292.9180 | 440.4685 | 391.1260 | 147.8300 | 257.5700 | 259.8085 | 149.7245 | 0 | 237.8790 |
| Hengshui | 299.7580 | 259.8555 | 142.5935 | 352.4360 | 508.4835 | 214.2995 | 167.0325 | 144.8405 | 455.2955 | 495.3555 | 140.0620 | 237.8790 | 0 |

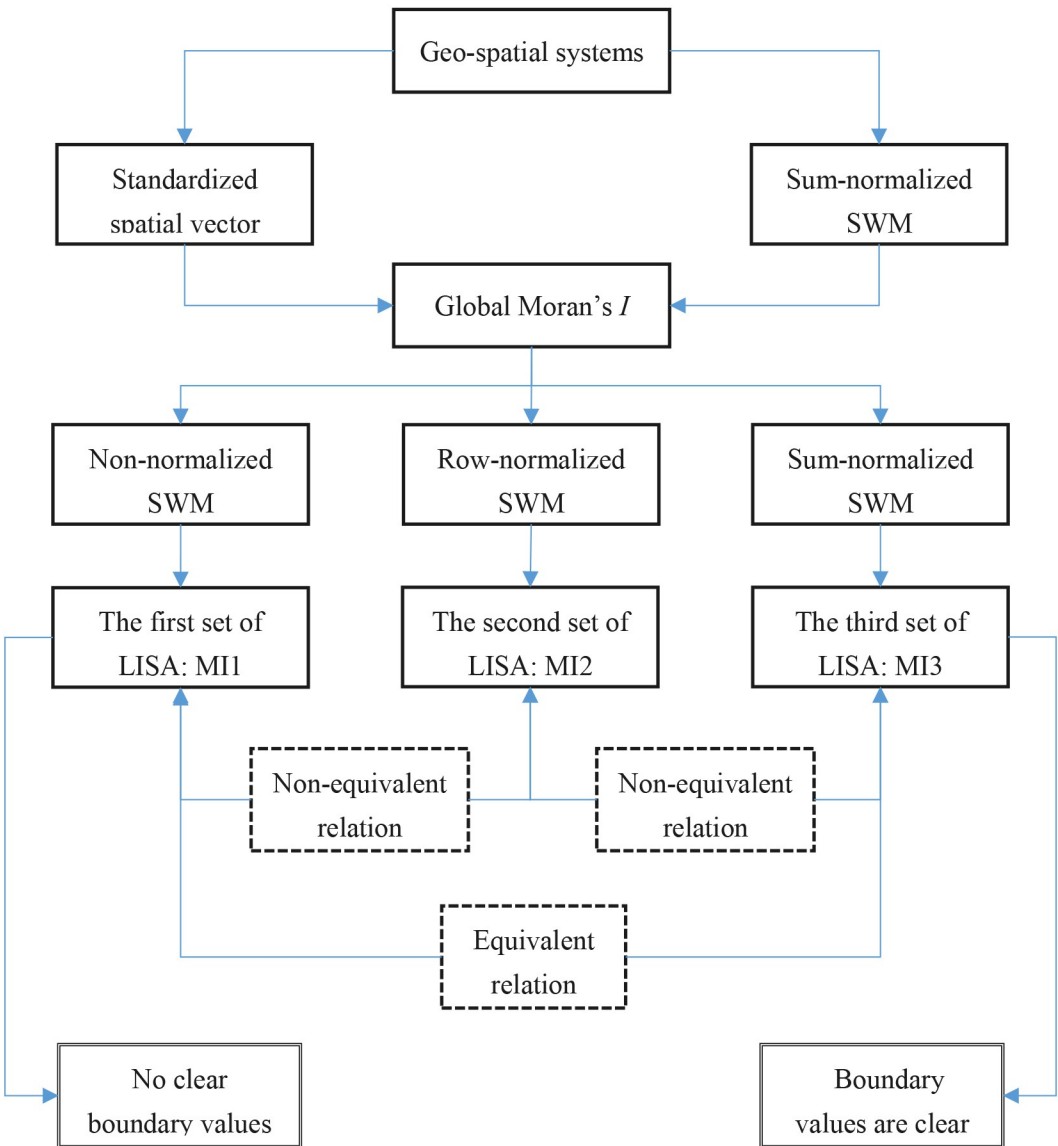

**Fig 1. A schematic flowchart of the conversion relationship from Moran's index to different types LISAs. (Note:**
Moran's index is taken as an example in this figure. By analogy, we can know the conversion process of the Geary's
coefficient. In fact, using Eqs (42) and (44), we can achieve the numerical conversion between Moran's index and Geary's
coefficient readily).

mileage matrix (**U**) can be transformed into a spatial contiguity matrix (**V**), which can be
changed to the global normalization weight matrix (**W**) and row normalization weight matrix
(**W\***).

## 3.2 Calculation results

For the data of two years and two statistics, i.e., local Moran index and local Geary coefficient,
three sets of calculation results are given, respectively. The calculation process is simple, easy
to understand, and the author's calculations can be repeated by readers using Microsoft Excel
(See S1 and S2 Datasets). For the local spatial statistics defined by Anselin [19], the first set of
local Moran index is expressed as MI1, the second set of local Moran index as MI2; the first set

**Table 6. Comparison of three sets of local Moran index values in two years.**

| City | 2000 | | | | | 2010 | | | | |
|------|------|------|------|------|------|------|------|------|------|------|
| | Local MI1 | Local MI2 | Local MI3 | MI1/MI2 | MI1/MI3 | Local MI1 | Local MI2 | Local MI3 | MI1/MI2 | MI1/MI3 |
| Beijing | -2686.4966 | -0.7067 | -0.0612 | 3801.3644 | 43916.8725 | -7140.4536 | -0.6690 | -0.0579 | 10673.67042 | 123312.1000 |
| Tianjin | -387.0133 | -0.0951 | -0.0088 | 4071.1117 | 43916.8725 | -1175.2192 | -0.1028 | -0.0095 | 11431.08104 | 123312.1000 |
| Shijiazhuang | -23.1481 | -0.0068 | -0.0005 | 3385.2705 | 43916.8725 | -14.4935 | -0.0015 | -0.0001 | 9505.340198 | 123312.1000 |
| Tanshan | -121.7919 | -0.0343 | -0.0028 | 3547.3310 | 43916.8725 | -603.5770 | -0.0606 | -0.0049 | 9960.382257 | 123312.1000 |
| Qinhuangdao | -142.9763 | -0.0607 | -0.0033 | 2356.2158 | 43916.8725 | -379.2385 | -0.0573 | -0.0031 | 6615.906335 | 123312.1000 |
| Handan | 170.5561 | 0.0533 | 0.0039 | 3202.3026 | 43916.8725 | 594.8129 | 0.0662 | 0.0048 | 8991.593275 | 123312.1000 |
| Xingtai | 185.0124 | 0.0511 | 0.0042 | 3618.1153 | 43916.8725 | 637.3519 | 0.0627 | 0.0052 | 10159.13409 | 123312.1000 |
| Baoding | -92.0058 | -0.0244 | -0.0021 | 3771.5181 | 43916.8725 | -335.7750 | -0.0317 | -0.0027 | 10589.86662 | 123312.1000 |
| Zhangjiakou | -231.9379 | -0.1057 | -0.0053 | 2194.2630 | 43916.8725 | -708.7104 | -0.1150 | -0.0057 | 6161.166944 | 123312.1000 |
| Chengde | -363.3994 | -0.1476 | -0.0083 | 2461.9446 | 43916.8725 | -889.9662 | -0.1287 | -0.0072 | 6912.777246 | 123312.1000 |
| Cangzhou | -194.7349 | -0.0538 | -0.0044 | 3620.4838 | 43916.8725 | -561.9455 | -0.0553 | -0.0046 | 10165.78443 | 123312.1000 |
| Langfang | -1369.3138 | -0.3073 | -0.0312 | 4455.7783 | 43916.8725 | -3399.6518 | -0.2717 | -0.0276 | 12511.16811 | 123312.1000 |
| Hengshui | 27.8793 | 0.0081 | 0.0006 | 3431.1735 | 43916.8725 | 120.3620 | 0.0125 | 0.0010 | 9634.229089 | 123312.1000 |
| Sum | -5229.3702 | -1.4299 | -0.1191 | 43916.8725 | 570919.3421 | -13856.5039 | -1.3523 | -0.1124 | 123312.1000 | 1603057.3005 |
| Expected | -5229.3702 | -1.5480 | -0.1191 | 43916.8725 | 570919.3421 | -13856.5039 | -1.4608 | -0.1124 | 123312.1000 | 1603057.3005 |

of local Geary coefficients is expressed as GC1, and the second set of local Geary coefficients is written as GC2. Accordingly, the modified local Moran index and Geary coefficient are expressed as MI3 and GC3, respectively (Fig 1). The results are as follows. First, the ratio of MI1 to MI2 is not a constant, and the ratio of GC1 to GC2 is also not a constant. This proves that the two sets of local Moran indices and the two sets of local Geary coefficients of Anselin [19] are not equivalent to one another; Secondly, the ratio of MI1 to MI3 is a constant, and the ratio of GC1 to GC3 is also a constant. It is proved that the first set of local Moran index of Anselin [19] is equivalent to the modified local Moran index in this paper, and the first set of local Geary coefficient of Anselin [19] is also equivalent to the modified local Geary coefficient of this paper (Tables 6 and 7). The reason is that the first set of local Moran index and local

**Table 7. Comparison of three sets of local Geary coefficient values in two years.**

| City | 2000 | | | | | 2010 | | | | |
|------|------|------|------|------|------|------|------|------|------|------|
| | Local GC1 | Local GC2 | Local GC3 | GC1/GC2 | GC1/GC3 | Local GC1 | Local GC2 | Local GC3 | GC1/GC2 | GC1/GC3 |
| Beijing | 41036.8054 | 10.7953 | 0.4313 | 3801.3644 | 95153.2237 | 113754.5272 | 10.6575 | 0.4258 | 10673.6704 | 267176.2168 |
| Tianjin | 12819.0307 | 3.1488 | 0.1347 | 4071.1117 | 95153.2237 | 37929.2182 | 3.3181 | 0.1420 | 11431.0810 | 267176.2168 |
| Shijiazhuang | 2908.7705 | 0.8592 | 0.0306 | 3385.2705 | 95153.2237 | 8029.3420 | 0.8447 | 0.0301 | 9505.3402 | 267176.2168 |
| Tanshan | 5340.6947 | 1.5056 | 0.0561 | 3547.3310 | 95153.2237 | 15962.5572 | 1.6026 | 0.0597 | 9960.3823 | 267176.2168 |
| Qinhuangdao | 3628.6681 | 1.5400 | 0.0381 | 2356.2158 | 95153.2237 | 10073.4191 | 1.5226 | 0.0377 | 6615.9063 | 267176.2168 |
| Handan | 2044.0978 | 0.6383 | 0.0215 | 3202.3026 | 95153.2237 | 5920.6445 | 0.6585 | 0.0222 | 8991.5933 | 267176.2168 |
| Xingtai | 2655.7337 | 0.7340 | 0.0279 | 3618.1153 | 95153.2237 | 7227.0101 | 0.7114 | 0.0270 | 10159.1341 | 267176.2168 |
| Baoding | 5080.6946 | 1.3471 | 0.0534 | 3771.5181 | 95153.2237 | 14731.9805 | 1.3911 | 0.0551 | 10589.8666 | 267176.2168 |
| Zhangjiakou | 4499.9163 | 2.0508 | 0.0473 | 2194.2630 | 95153.2237 | 12851.4607 | 2.0859 | 0.0481 | 6161.1669 | 267176.2168 |
| Chengde | 5353.0964 | 2.1743 | 0.0563 | 2461.9446 | 95153.2237 | 14332.0819 | 2.0733 | 0.0536 | 6912.7772 | 267176.2168 |
| Cangzhou | 5400.0965 | 1.4915 | 0.0568 | 3620.4838 | 95153.2237 | 15101.1057 | 1.4855 | 0.0565 | 10165.7844 | 267176.2168 |
| Langfang | 13324.4547 | 2.9904 | 0.1400 | 4455.7783 | 95153.2237 | 35822.5797 | 2.8632 | 0.1341 | 12511.1681 | 267176.2168 |
| Hengshui | 4161.8231 | 1.2129 | 0.0437 | 3431.1735 | 95153.2237 | 10946.6401 | 1.1362 | 0.0410 | 9634.2291 | 267176.2168 |
| Sum | 108253.8824 | 30.4883 | 1.1377 | 43916.8725 | 1236991.9079 | 302682.5671 | 30.3506 | 1.1329 | 123312.1000 | 3473290.8178 |
| Expected | 108253.8824 | 32.0446 | 1.1377 | 43916.8725 | 1236991.9079 | 302682.5671 | 31.9099 | 1.1329 | 123312.1000 | 3473290.8178 |

Geary coefficient defined by Anselin [19] are based on symmetric spatial contiguity matrix. The modified statistics in this paper are based on the globally normalized spatial weight matrix which is symmetric, while second set of local Moran index and local Geary coefficient defined by Anselin [19] are based on the locally normalized spatial weight matrix, in which the symmetry is broken.

Using the calculation results, we can verify two key equations. The relationship between the sum of the first set of local Moran indexes and the global Moran index satisfies Eq (8), and the relationship between the sum of the first set of local Geary coefficients and the global Geary coefficient satisfies Eq (26). However, the relationship between the sum of the second set of local Moran indexes and the global Moran index does no satisfy Eq (16), and the relationship between the sum of the second set of local Geary coefficients and the global Geary coefficient does not satisfy Eq (27). The sum of spatial contiguity matrices is $V_0 = 0.6671$. In 2000, the population variance of city population in Beijing-Tianjin-Hebei region is $\sigma^2 = 65835.5974$, thus $\gamma = \sigma^2 V_0 = 43916.8725$, the global Moran index is $I = -0.1191$, and the sum of the first set of local Moran indexes is $\Sigma I_i^* = -5229.3702 = \gamma I = 43916.8725*(-0.1191)$. On the other hand, $n = 13$, $\gamma_c = 2nV_0/(n-1) = 1.4453$, and the global Geary coefficient is $C = 1.1377$, so the sum of the first set of local Geary coefficients is $\Sigma C_i^* = 108253.8824 = \gamma_c\sigma^2 C = 1.4453*65835.5974*1.1377$. However, the sum of the second set of local Moran indices is $\Sigma I_i^{**} = -1.4299$, while $n*I = 13*(-0.1191) = -1.5480$. The two values are not equal to one another ($-1.4299 \neq -1.5480$). The sum of the second set of local Geary coefficients is $\Sigma C_i^{**} = 30.4883$, and $2n^{2*}C/(n-1) = 28.1667*1.1377 = 32.0446$. The two values are not equal to one another ($30.4883 \neq 32.0446$). These results indicate that, based on the conventional formula for the second sets of LISA, Anselin's [19] second basic requirement cannot be met. The sum of the third set of local Moran index is equal to the global Moran index, the ratio of the first set of local Moran indexes to the corresponding third set of local Moran indexes is $\gamma = \sigma^2 V_0 = 43916.8725$, which is a constant; the sum of the third set of local Geary coefficients equals the global Geary coefficient, and the ratio of the first set of local Geary coefficients to the corresponding third set of local Geary coefficient is $\gamma_c\sigma^2 = 1.4453* 65835.5974 = 95153.2237$ is a constant (Tables 6 and 7). This suggests that, based on improved formulae, Anselin's [19] second basic requirement can be met by the calculation results.

The calculation result of one year may be regarded as an isolated case, so we might as well take a look at the situation in 2010. Based on the 6th census data, the population variance of Beijing-Tianjin-Hebei city population is $\sigma^2 = 184856.6464$, thus $\gamma = \sigma^2 V_0 = 123312.1000$, the global Moran index is $I = -0.1124$, and the sum of the first set of local Moran indexes is $\Sigma I_i^* = -13856.5039 = \gamma I = 123312.1000*(-0.1124)$. On the other hand, $\gamma_c = 1.4453$, and the global Geary coefficient is $C = 1.1329$, so the sum of the first set of local Geary coefficients is $\Sigma C_i^* = 302682.5671 = \gamma_c\sigma^2 C = 1.4453*184856.6464*1.1329$. However, the sum of the second set of local Moran indices is $\Sigma I_i^{**} = -1.3523$, while $n*I = 13*(-0.1124) = -1.4608$ (Fig 2(A)). The two numbers are not equal to each other ($-1.3523 \neq -1.4608$). The sum of the second set of local Geary coefficients is $\Sigma C_i^{**} = 30.3506$, and $2n^{2*}C/(n-1) = 28.1667*1.1329 = 31.9099$. The two numbers are not equal to each other ($30.3506 \neq 31.9099$). These results once again indicate that Anselin's [19] second basic requirement cannot be satisfied through common formula. The sum of the third set of local Moran index is equal to the global Moran index, the ratio of the first set of local Moran indexes to the corresponding numbers in the third set of local Moran index is $\gamma = \sigma^2 V_0 = 123312.1000$ (Fig 2(B)); the sum of the third set of local Geary coefficients equals the global Geary coefficient, and the ratio of the first set of local Geary coefficient to the corresponding third set of local Geary coefficient is $\gamma_c\sigma^2 = 1.4453* 184856.6464 = 267176.2168$ is a constant (Tables 6 and 7). This suggests that, based on new formulae, Anselin's [19] second basic requirement can be satisfied once again by the calculation results. It can be seen that the

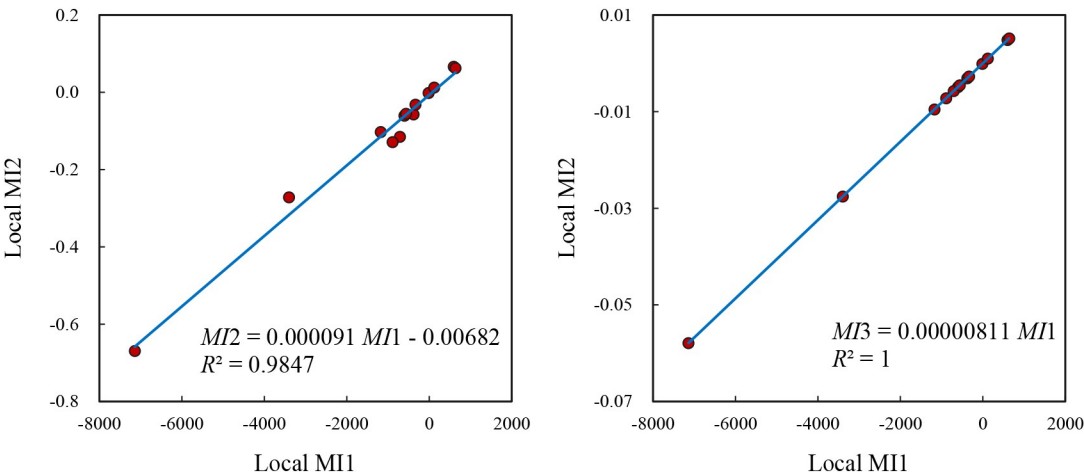

**Fig 2. The relationships between three sets of local Moran's indexes of BTH cities in 2010.** (a) MI2 vs MI1 (high correlation). (b) 2MI3 vs MI1 (perfect fit) (Note: The second set of local Moran's indexes (MI2) are highly correlated with the first local Moran's indexes (MI1), but not equivalent to one another. The third set of local Moran's indexes (MI3) is equivalent to the first set of local Moran's indexes (MI1). The coefficient $1/\gamma = 1/123312.1000 = 0.000008110$. MI2 does not satisfy the second requirement for LISAs given by Anselin [19]).

calculation results of the two years fully support the previous theoretical inferences and related judgments.

## 4 Questions and discussion

The re-expressed local Moran indexes and the local Geary coefficients in this work are derived from Anselin's correct definition and relationship, without substantial innovation. The contribution of this study lies in three aspects. First, it clarifies a series of logical misunderstandings of local spatial autocorrelation statistics and gives the correct expressions. Second, it normalizes the local spatial autocorrelation statistics, and the canonical results are helpful for more convenient application. Third, it clarifies a number of fundamental concepts related to spatial autocorrelation that have long been confused in literature. In terms of the tradition of statistics, important concepts and their symbols have been distinguished. Especially, it emphasizes the distance axiom hidden behind the spatial weight matrix. If the spatial contiguity matrix is normalized by row, the locally normalized spatial weight matrix will bear a different mathematical structure from the non-normalized spatial weight matrix and the globally normalized spatial weight matrix by sum. Applying the results derived from the models based on non-normalized spatial weight matrix to the relation formulae based on row-normalized spatial weight matrix results in wrong mathematical expressions. Generally speaking, spatial contiguity matrix is of symmetry. Therefore, non-normalized spatial weight matrix and globally normalized spatial weight matrix are symmetric. Substitution of symmetric spatial weight matrix with asymmetric spatial weight matrix leads to two wrong relations: First, the sum of local Moran index based on standardized variable and local normalized weight matrix is equal to $n$ times of global Moran index; Second, the sum of local Geary coefficients based on standardized variable and local normalized weight matrix is equal to $2n^2/(n-1)$ times of global Geary coefficient. In fact, the two relations can never be derived from Anselin's original assumptions.

The errors based on the wrong relations are not too significant in many cases, but the results have a far-reaching impact on geographical analysis. Concretely speaking, these incorrect relationships lead to a series of problems (Table 8): (1) The relationship between the

**Table 8. Functions and problems of Anselin's LISA and the improved effect of this paper.**

| Definer | Variable | Statistic | Function | Advantages and disadvantages |
|---|---|---|---|---|
| **Anselin** | Central variable and non-normalized symmetric contiguity matrix | First local Moran's $I$ | Reflect local spatial dependence | Simple but lack of clear boundary value and critical value (reference value) |
| | | First local Geary's $C$ | Reflect local spatial dependence | Simple but lack of clear boundary value and critical value (reference value) |
| | Standard variable and row-normalized asymmetric weight matrix | Second Moran's $I$ | Reflect local spatial dependence from the perspective of population | Decoupled from the first definition of local Moran's $I$; Decoupling from correlation coefficient; The relationships between two elements in the system is ignored |
| | | Second Geary's $C$ | Reflect local spatial dependence from the perspective of population | Decoupled from the first definition of local Geary's $C$; Decoupling from the analogy with the Durbin-Watson statistic; The relationships between two elements in the system is ignored; sample standard deviation is replaced by population standard deviation |
| **This paper** | Standardized variable and global normalized symmetric weight matrix | Third Moran's $I$ | Reflect local spatial dependence from the perspective of population | Equivalent to the first definition of local Moran's $I$; Linked to correlation coefficient; The spatial relationship of other elements other than the target geographical elements is considered; There are clear boundary values and critical values |
| | | Third Geary's $C$ | Reflect local spatial dependence from the perspective of samples | Equivalent to the first definition of local Geary's $C$; Linked to generalized Durbin-Watson statistics; The spatial relationship of other elements other than the target geographical elements is considered; Return to the sample analysis perspective of global Geary coefficient; There are clear boundary values and critical values |

definitions of two local Moran indexes is broken (not equivalent to each other). The first set of local LISA is based on symmetric spatial adjacency matrix, and the second set is based on asymmetric spatial weight matrix normalized by row. As a result, the ratio of the values of the two sets of parameters is not a constant. (2) When defining the local spatial autocorrelation index, we only consider the relationship between one element and other elements. The pairwise correlation between all elements is ignored. That is, for the local index of the $i$th geographical element, only the relationships between element $i$ and element $j$ are taken into account, the relationships between element $j$ and element $k$ are neglected ($i$, $j$, $k$ = 1,2,3,...,$n$). In this case, the wholeness of a geographical system is overlooked in the local spatial analysis. (3) The absolute value of the local Moran index may exceed 1, thus decoupling from the concept of correlation coefficient. Moran's index was proposed by analogy with Pearson correlation. The values of Moran's index comes between -1 and 1. (4) The parameters are lack of clear boundary value and critical value. The absolute boundary values of Moran index is -1 and 1. The critical value is 0 in theory and $1/(1-n)$ in experience. The boundary values of the Geary coefficient are 0 and 2, and the critical value is theoretically 1. In addition, Anselin [19] used the population standard deviation to replace the sample standard deviation when defining the

local Geary coefficient. Where logic is concerned, no problem; while where history is concerned, there is problem: the result violates the original intention of the definition of Geary coefficient. In spatial analysis, it is sometimes difficult to distinguish between spatial samples and spatial populations. Moran's index, which is derived from Pearson correlation coefficient, as indicated above, is a statistics based on population standard deviation. Geary's coefficient is defined by analogy with Durbin-Watson statistics based on sample standard deviation in order to make up for the deficiency of Moran's index. To define the local Geary coefficient, we should respect the original meaning of the definition of the Geary coefficient, so that the local Geary coefficient can be effectively associated with the global Geary coefficient. From the existing literature, some readers have found Anselin's mistakes. Some scholars adopt a compromise approach. For example, they use the global normalized spatial weight matrix instead of the local normalized spatial weight matrix by row, but multiply $n$ in front of the corrected local Moran index calculation formula—I found this kind of treatment in some teaching courseware. This ensures that the sum of local Moran indexes is equal to $n$ times the global Moran index.

As we know, Anselin is a well-known outstanding scholar in the field of geographical spatial analysis. Due to the far-reaching influence of Anselin's work, its logical errors caused confusion in its application and interpretation. Science respects logic and facts, not authority—only pseudoscience starts from authoritative judgment. In order to solve the above problems, this paper carries out the following processing in the process of mathematical deduction: First, return to the essence of the spatial distance matrix behind the spatial weight matrix, and respect the basic distance axiom. The global spatial weight matrix is obtained by global normalization of spatial contiguity matrix. The globally normalized spatial weight matrix is used to replace Anselin's row-normalized weight matrix. In this way, the connotation of the concept before and after is unified and the logic is consistent, so as to avoid reasoning mistakes. Second, start from the original idea of Moran's index and Geary's coefficient. The normalized local Moran's index is defined, and the population standard deviation is used to standardize the size variable; the normalized local Geary's coefficient is defined, and the sample standard deviation is used to standardize the size variable. Third, start from the original intention of Anselin [19]. Anselin gives two sets of local Moran's index and local Geary's coefficient. But there is inconsistency between them. By examining the reasoning process, we can find that the reason for the error lies in the logic error caused by the unintentional concept replacement. According to the sign system and simplification principle of this paper, we transform Anselin's second set of local Moran index and local Geary coefficient formulae. Comparing the two sets of results, we can see the problems and thus understand the similarities and differences between the two sets of formulae (Tables 8 and 9).

Finally, it is appropriate to briefly discuss the definition of spatial weight matrix. Spatial autocorrelation analysis depends on spatial contiguity matrix, which has multiple definitions. In fact, definition of spatial contiguity involves different spatial effects. Spatial effects of geographical processes fall into two categories: *action at a distance* and *local action* [37]. Local action can be expressed with step function in mathematics and nominal variable in value. In spatial autocorrelation analysis, the spatial contiguity matrix based on local action is mainly applicable to relationships between regions. The spatial contiguity relationship of regions bears three ways of definitions, that is, Rook's contiguity, Bishop's contiguity, and Queen's contiguity [38]. Rook's contiguity plus Bishop's contiguity yields Queen's contiguity. In fact, Rook's contiguity corresponds to von Neumann's neighborhood definition, while Queen's contiguity corresponds to Moore's neighborhood definition [39]. Action at a distance can be reflected by certain distance, including Euclidean distance, travel time, transportation mileage and so on. When converting distances into spatial contiguity matrix, a certain spatial

**Table 9. Comparison of between normalized LISA and the equivalent transformation results of Anselin's second set of LISA definitions.**

| Category | Measure | Definition in this paper | Anselin's definition |
|---|---|---|---|
| Moran's $I$ | Global Moran's $I$ | $I = \sum_{i=1}^{n} \sum_{j=1}^{n} w_{ij} z_i z_j = \mathbf{z}^{\mathrm{T}} \mathbf{W} z$ | $I = \sum_{i=1}^{n} \sum_{j=1}^{n} w_{ij} z_i z_j$ |
| | Local Moran's $I$ | $I_i = z_i \sum_{j=1}^{n} w_{ij} z_j$ | $I_i = \frac{z_i}{V_i} \sum_{j=1}^{n} v_{ij} z_j$ |
| | Sum of local Moran's I | $\sum_{i=1}^{n} I_i = I$ | $\sum_{i=1}^{n} I_i \approx nI$ |
| Geary's $C$ | Global Geary's $C$ | $C = \frac{1}{2} \sum_{i=1}^{n} \sum_{j=1}^{n} w_{ij} (z_i^* - z_j^*)^2$ | $C = \frac{1}{2} \sum_{i=1}^{n} \sum_{j=1}^{n} w_{ij} (z_i^* - z_j^*)^2$ |
| | Local Geary's $C$ | $C_i = \frac{1}{2} \sum_{j=1}^{n} w_{ij} (z_i^* - z_j^*)^2 \quad = \frac{n-1}{2n} \left( \sum_{j=1}^{n} w_{ij} (z_i^2 + z_j^2) \right) - 21$ | $C_i = \frac{1}{V_i} \sum_{j=1}^{n} v_{ij} (z_i - z_j)$ |
| | Sum of Local Geary's $C$ | $\sum_{i=1}^{n} C_i = C = \frac{n-1}{n} (\mathbf{e}^{\mathrm{T}} \mathbf{W} \mathbf{z}^2 - I)$ | $\sum_{i=1}^{n} C_i \approx \frac{2n^2}{n-1} C$ |

**Note**: For comparison, Anselin's definitions are transformed and re-expressed with new symbols. However, the new expressions are completely equivalent to Anselin's original expressions.

contiguity function needs to be adopted. Common spatial contiguity functions include absolute step function, relative step function, exponential function, and distance inverse function (a type of hyperbolic function) [6, 12, 27]. Distance-based spatial contiguity matrix is suitable for networks of locations such as urban systems. In this case, based on the step function, spatial contiguity is represented by nominal variable (dummy variable in discrete format); based on other functions, the spatial contiguity is represented by metric variable (continuous variable). Although the function expressions are different, the logic behind them is consistent with one another. Mathematics is the pinnacle of logic. In mathematics, the most basic function is exponential function. Various forms of simple functions can be reduced to exponential function. The step function is an extreme form of an exponential function, and moving average on the step function can yield an inverse distance function [40]. So, using different functions to define spatial contiguity matrices will definitely affect the calculation results, but it has no impact on the mathematical reasoning results and the logical relationships behind them. The reason why row normalization weight matrix affects mathematical reasoning results is because the logic behind the spatial weight matrix has been changed, and the logic is regulated by the distance axiom. Scientific research typically involves three worlds: the *real world*, the *mathematical world*, and the *computational world* [41]. The process of mathematical transformation and derivation belongs to the mathematical world, while the selection of spatial weight matrix forms belongs to the computational world. The key is to choose the appropriate spatial contiguity matrix definition method for different geographic systems based on different situations [27]. One obvious drawback of this study is the lack of empirical analysis based on different types of spatial weight matrices. Therefore, the influence of types and structure of spatial contiguity matrixes on theoretical modelling and computational results of spatial autocorrelation appears hollow.

## 5 Conclusions

The global spatial autocorrelation coefficients reflect the sum of any two geographical elements in a region, while the local spatial autocorrelation indexes reflect the sum of correlation

between a geographical element and all other geographical elements. The sum of parts is proportional to the whole. The first set of local Moran indexes and Geary coefficients defined by Anselin [19] is effective and consistent with the idea of global Moran index and Geary coefficient. However, the second set of local Moran indexes and local Geary coefficients defined by him are not equivalent to the first set of parameters. The non-normalized spatial weight matrix is isomorphic to the sum-based normalized spatial weight matrix, but not isomorphic to the row-based normalized spatial weight matrix. The derived results based on non-normalized spatial weight matrix cannot be directly applied to the mathematical relations based on row-normalized spatial weight matrix. The key issue rests that Anselin [19] directly applied the derived results based on the non-normalized spatial weight matrix to the relationship formula based on the row-normalized spatial weight matrix. This paper is devoted to correcting the unintentional mistakes in his reasoning process and gives the third set of definitions of local Moran indexes and local Geary coefficient in canonical forms. The newly-defined local Moran index and local Geary coefficient are simple and concise. The improved expressions are consistent with the original intention of Anselin [19] and the statistical essence of global Moran index and global Geary coefficient.

Local spatial autocorrelation analysis is a methodology developed on the basis of global spatial autocorrelation analysis. The progress of science has no end. The main points of this paper are summarized as follows. Firstly, the LISA defined in literature is of great significance for analysis of local spatial autocorrelation, but there are also some faults. The first set of LISA is based on the definition of centralized variables and non-normalized spatial contiguity matrix, lacking clear boundary values and critical value. The second set of local LISA is based on the definitions of standardized variables and row-normalized spatial weight matrix, which ignores the global relationship behind the local analysis. One of the results is that the two sets of indexes are not equivalent to one another. In addition, the population standard deviation is adopted when defining the second local Geary coefficients, which violates the original intention of Geary coefficient. All the indexes lack clear boundary values and critical value, and they are uncoupled from the correlation coefficient. One consequence is that the analysis process is complex; the other is that the conclusions drawn from the two sets of indexes are often inconsistent with each other. Secondly, the LISA expression is reconstructed by using the sum-normalized spatial weight matrix and standardized size variables based on *z*-score to eliminate the defects of Anselin's LISA definition. By doing so, we have canonical spatial autocorrelation measurements. The sum-based globally normalized spatial weight matrix is used to replace the row-based locally normalized spatial weight matrix. The population standard deviation is used to standardize the variables when defining the local Moran indexes, and the sample standard deviation is used to standardize the variables when defining the local Geary coefficient. The local LISA problem of Anselin [19] can be solved effectively and the results are more concise and simpler. The results given in this paper are equivalent to those given by Anselin's first set of formulas, i.e. first sets of local Moran index and local Geary coefficient, but they are not linearly proportional to the results of the second set of formulas, namely the second sets of local Moran index and local Geary coefficient.

## Supporting information

**S1 File. Anselin's derivation and expressions for LISA.** This is a microcosm of Anselin's paper on LISA. The key parts of Anselin's mathematical reasoning are extracted, and the main errors in the reasoning process are revealed. This file uses Anselin's original symbol system. Through this file, readers can more easily grasp the essence of the problem.
(DOCX)

**S2 File. Value transformation methods and formulae.** This file show common concepts and methods of value transformation and corresponding formulae for variable standardization. This document clarifies some confusion and inappropriate expressions regarding variable standardization in the literature.
(DOCX)

**S1 Dataset. Spatial data sets and calculation results of local spatial autocorrelation indexes for 2000.** This file includes the dataset of spatial distances and city population in 2000, global Moran's indexes and Geary's coefficients, three sets of local Moran's index, and three sets of local Geary's coefficients. The original data and calculation process are displayed for readers.
(XLSX)

**S2 Dataset. Spatial data sets and calculation results of local spatial autocorrelation indexes for 2010.** This file includes the dataset of spatial distances and city population in 2010, global Moran's indexes and Geary's coefficients, three sets of local Moran's index, and three sets of local Geary's coefficients. All the results are tabulated for comparison and references.
(XLSX)

## Acknowledgments

My student, Dr. Yuqing Long, has extracted spatial distance matrix data from the Beijing Tianjin Hebei urban network map for me, and I would like to express my gratitude. I would like to thank the anonymous reviewer and Dr. Yuxia Wang whose interesting and constructive comments were very helpful in improving the quality of this paper. The academic editor, Dr. Yuxia Wang, put in tremendous effort to invite reviewers for this paper, and I am particularly grateful for it.

## Author Contributions

**Conceptualization:** Yanguang Chen.

**Data curation:** Yanguang Chen.

**Formal analysis:** Yanguang Chen.

**Funding acquisition:** Yanguang Chen.

**Investigation:** Yanguang Chen.

**Methodology:** Yanguang Chen.

**Project administration:** Yanguang Chen.

**Resources:** Yanguang Chen.

**Software:** Yanguang Chen.

**Supervision:** Yanguang Chen.

**Validation:** Yanguang Chen.

**Visualization:** Yanguang Chen.

**Writing – original draft:** Yanguang Chen.

**Writing – review & editing:** Yanguang Chen.

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
