## [Decision Letter · Decision Letter 0]

21 Feb 2024

PONE-D-23-35394Reconstruction and Normalization of LISA for Spatial AnalysisPLOS ONE

Dear Dr. Chen,

Thank you for submitting your manuscript to PLOS ONE. After careful consideration, we feel that it has merit but does not fully meet PLOS ONE’s publication criteria as it currently stands. Therefore, we invite you to submit a revised version of the manuscript that addresses the points raised during the review process.  

Dear authors, I continued inviting around 30 reviewers but only received one comments. To ensure a timely review, I served as another reviewer. Please the suggestions and comments.

We look forward to receiving your revised manuscript.

Kind regards,

Yuxia Wang

Academic Editor

PLOS ONE

[This research was sponsored by the National Natural Science Foundation of China (Grant No. 42171192). The support is gratefully acknowledged.]

 [The author(s) received no specific funding for this work.]

Additional Editor Comments:

The authors conduct a series of rigorous mathematical reasoning of LISA showing that using row-normalized spatial weight matrix would violate the second basic requirement for LISA. As stated by the authors, this is not substantial innovation, but it is helpful in figuring the logic of local spatial autocorrelation statistics. I have some minor comments.

1. Page 4. The spatial contiguity matrix V is not explained in detail. There are many definitions of spatial contiguity matrix, and would Rook, Queen, or distance-based matrix have any difference on the calculation of LISA?

2. Page 4. Is it necessary to stress that i≠j in Equation (1)?

3. Page 4 to Page 5. It might be that I misunderstand something. The z-score normalization is only divided by σ, what is the meaning of σ. It seems that σ is calculate by x. But if we treat x_i-x ® as a whole, the normalization should be based on the σ of x_i-x ®. Compared with equation 1, dividing only by σ. could not be called z-score normalization.

4. Page 6 Table 1. What is the benefit of using the normalized weight matrix instead of the original one?

5. Page 2 In the first paragraph of introduction, “Gravity models, spatial interaction models, and spatial autocorrelation analysis are the main approaches…”, parallel relationship might not be appropriate for gravity model and spatial interaction model since the former is one type of the latter.

Reviewers' comments:

Reviewer's Responses to Questions

**Comments to the Author**

1. Is the manuscript technically sound, and do the data support the conclusions?

Reviewer #1: Yes

2. Has the statistical analysis been performed appropriately and rigorously? 

Reviewer #1: Yes

3. Have the authors made all data underlying the findings in their manuscript fully available?

Reviewer #1: Yes

4. Is the manuscript presented in an intelligible fashion and written in standard English?

Reviewer #1: Yes

5. Review Comments to the Author

Reviewer #1: The article is an interesting work. Provide clarifications on the local spatial association indicator (LISA) and the local Geary indicator, widely used in the literature. This article aims to reconstruct the calculation formulas of local Moran indices and Geary coefficients through mathematics, presenting corrections or modifications to these indicators. Finally, it presents an application to real data.

6. PLOS authors have the option to publish the peer review history of their article (what does this mean?). If published, this will include your full peer review and any attached files.

Reviewer #1: No

---

## [Author Response · Author response to Decision Letter 0]

11 Apr 2024

Please see the attached file entitled "Response to Reviewers"

---

## [Editor Report · Decision Letter 1]

25 Apr 2024

Reconstruction and Normalization of LISA for Spatial Analysis

PONE-D-23-35394R1

Dear Dr. Chen,

We’re pleased to inform you that your manuscript has been judged scientifically suitable for publication and will be formally accepted for publication once it meets all outstanding technical requirements.

Kind regards,

Yuxia Wang

Academic Editor

PLOS ONE
---

## [Editor Report · Acceptance letter]

10 May 2024

PONE-D-23-35394R1 

PLOS ONE

Dear Dr. Chen, 

I'm pleased to inform you that your manuscript has been deemed suitable for publication in PLOS ONE. Congratulations! Your manuscript is now being handed over to our production team.

Kind regards, 

on behalf of

Dr. Yuxia Wang 

Academic Editor

PLOS ONE